

# Trp-574-Leu mutation in wild mustard (*Sinapis* arvensis L.) as a result of als inhibiting herbicide applications

Bahadir Sin and İzzet Kadıoglu

Department of Plant Pathology, Faculty of Agriculture, Tokat Gaziosmanpasa University, Tokat, Turkey

## ABSTRACT

Wheat is one of the most important crops grown all around the world. Weeds cause significant yield loss and damage to wheat and their control is generally based on herbicide application. Regular use leads to herbicide resistance in weeds. This study aims to reveal molecular detection of *Sinapis arvensis* resistance mutation against ALS inhibiting herbicides. For this purpose, survey studies have been carried out in wheat growing areas in Amasya, Çorum, Tokat, and Yozgat provinces and wild mustard seeds have been collected from 310 different fields. According to bioassay tests with tribenuron-methyl, 13 of these populations, have not been affected by the registered dose of herbicide. When survived populations were subjected to dose-effect study and herbicides were applied at 26-fold, the highest and lowest resistance coefficients were determined as 7.2 (A-007) and 1.02 (T-034) respectively. In addition, B domain region from ALS gene was amplified and analyzed in molecular studies to determine point mutation in wild mustard against ALS herbicides. The PCR products were sequenced and target-site mutation to Leucine was observed at Trp-574 amino acide. In the study, point mutation in Trp-574 amino acide and Trp-574 Leu mutation in *Sinapis arvensis* have been detected for the first time in Turkey.

# INTRODUCTION

Wheat is an important, high-fiber food consumed daily by human beings. Approximately 21% of one's daily caloric needs are supplied by wheat products (*Shiferaw et al., 2013*). Cereal grains provide 52% of the total protein and 50% of the total carbohydrate and energy worldwide; in Turkey specifically they account for 60% and 80%, respectively. Wheat consumption is higher in poor countries (*Atak, 2004*; *Aydın et al., 2005*). Food demand parallels rapid population growth and results in the enlargement of wheat production areas and, consequentially, an increased yield. The goals of production are to produce a higher quality product and yield from areas with restricted agricultural capacity around the world.

The reduction in yield and quality are major problems in cultivated areas. Cultivated crops are impacted by ecological conditions, stress factors, and plant pests (insects, spider mites, nematodes, diseases, and weeds) and may encounter significant yield losses. Weeds, pests, and pathogens yield worldwide crop losses estimated to be 34%, 18%, and 16%, respectively. Overall economic crop losses due to weeds have cost 75.6 billion U.S. dollars,

Corresponding author
Bahadir Sin, bahadir.sin@gop.edu.tr

which is equivalent to 2.5–2.7 billion dollars in Australia and five billion dollars in the USA (*Schneider, 1985*; *Combellak, 1987*; *Oerke et al., 1994*; *Jabran et al., 2015*).

Weeds are a major cause of yield loss in wheat. Weeds are capable of germinating before wheat typically emerges and grows, competing with the crop plant for space, nutrients, water, and sunlight (*Özer et al., 2003*; *Oerke & Dehne, 2004*). Weeds may cause critical damage to wheat approximately 30-60 days after sowing (*Marwat et al., 2013*). Apart from these direct and indirect damages, weeds may also reduce the wheat quality and seed value by mixing with their seeds to products and cause color, taste, and odor deterioration when they are milled with flour. Several pests and disease agents hosted by weeds may cause secondary infections (*Özer et al., 2003*; *Bülbül & Ve Aksoy, 2005*; *Anonymous, 2008*; *Güncan & Ve Karaca, 2014*).

The average yield loss to wheat is estimated to be 15.6%, despite weed control measures (*Oerke et al., 1999*; *Oerke, 2006*; *Güncan & Ve Karaca, 2014*). Yield loss reaches up to 100% in non-controlled fields and yield loss is reported to be 20–35% in Turkey (*Güncan, 1972*; *Anonymous, 2008*). Wheat is not hoed and weed control is typically conducted using chemicals, the specific amount of phytotoxic active substances depending on the type of herbicide. These chemicals cause death or reduce growth of weeds by inhibiting certain reactions such as amino acid synthesis, photosynthesis, and lipid biosynthesis (*Ghorbani, Leifert & Seel, 2005*). In 2020, 3.5 million tons of pesticides were used across the globe. Herbicides constitute approximately 50% of the pesticides used in 2010 (*Sharma et al., 2019*; *FAO, 2020*).

Herbicide application increases daily and herbicide resistance may occur with improper use. Herbicide resistance has developed in 262 plants around the world (152 dicotyledon, 110 monocotyledon). A total of 512 cases have been reported in 23 of 26 known herbicide groups in 92 plants in 70 countries (*Corbett & Tardif, 2006*; *Heap, 2014*; *Heap, 2020*).

Resistance was confirmed for the first time in *Senecio vulgaris* in 1957 (*Holt, 1988*). ALS herbicides have been used heavily since they were commercialized in 1980 (*Mallory-Smith, Thill & Dial, 1990*). The first report of ALS resistance occurred 1982 in *Lolium rigidum* in Australia. However, the first case in tribenuron-methyl resistance in wild mustard was reported in canola fields in 2002 in Canada (*Heap, 2020*).

ALS-inhibiting herbicides play a significant role in weed control for wheat and are preferred due to their efficacy in lower doses and lower toxicity to mammals (*De Prado & Franco, 2004*). Herbicide resistance was observed in corn fields in 1983 in *Sinapis arvensis* against the photosystem II inhibitor (C1) group herbicide, atrazine. ALS herbicide resistance was found in winter barley, wheat, and canola, and tribenuron-methyl resistance was discovered in canola and cereals in 2002 (*Ali, McLaren & Souza Machado, 1986*; *Heap & Morrison, 1992*; *Debreuil, Friesen & Morrison, 1996*; *Morrison & Devine, 1994*; *Warwick, Sauder & Beckie, 2005*; *Heap, 2020*). Eighteen cases in 15 herbicide-weed actions have been reported in Turkey. Five of those cases were resistant to ALS-inhibiting herbicides, six showed multiple resistances, and two were resistant in *S. arvensis* (*Heap, 2020*).

The continuous use of ALS herbicides promotes the rapid emergence of resistance in weeds. There are several mechanisms for the development of resistance, however target site mutation is considered to be the most important of these mechanisms (*Devine & Shukla,*

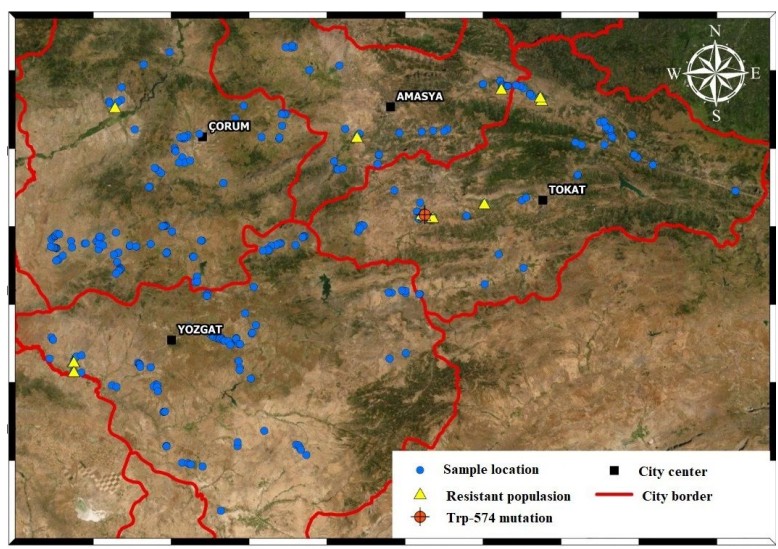

**Figure 1** **Sampled areas map.** Each of the blue dots indicates the points where wild mustard samples were collected. Yellow dots are resistant sample and orange dot is TRP-574 mutation population.

*2001*; *Grassel, 2002*). Herbicides cannot bind to their target site and weeds become resistant as a result of alterations in amino acids caused by mutations (*Kaya-Altop, Mennan & Işık, 2017*). Mutations are seen in the amino acid locations Ala 122, Ala 205, Asp 376, Gly 654, Pro 197, Trp 574, and Ser 653 (*Powles & Yu, 2010*; *Vencill et al., 2012*). These locations may vary depending on the herbicides' active reagent.

We sought to investigate target site mutations in the ALS gene in ALS-resistant *S. arvensis* found in wheat fields.

## MATERIALS & METHODS

### Collection of plant material
We used wild mustard seeds collected in April-July, 2018 from wheat fields in Amasya, Tokat, Çorum, and Yozgat. There were 36, 62, 109 and 103 populations in the four provinces, respectively. We used a control population obtained from a field in Tokat where no herbicides were applied. A total of 311 populations, including the control, were collected during our field surveys (Fig. 1).

### Resistance determination assay
The level of ALS herbicide resistance of the populations was determined by a single-dose assay experiment arranged by *Moss et al. (1999)*. All populations were treated with 10 g a.i. ha$^{-1}$ tribenuron-methyl and were screened for 1 month. The assay test results showed that populations with less than 80% reagent efficacy were putative resistant populations and others with higher efficacy were susceptible.

## Dose–responses to tribenuron-methyl

The undetermined populations selected from the previous resistance assay were treated with the registered dose of herbicide at one, two, four, eight and 16-fold levels. Susceptible populations were treated with 0.25 (2.5 g a.i. ha$^{-1}$) and 0.5 (5 g a.i. ha$^{-1}$) doses and the assay was established in 1.2 kg pots. The trials were conducted with four replicates with two repeats for each population and herbicide dose. Each pot was covered with polyethylene and maintained in greenhouse conditions after spraying. The length of the experiment was determined and trials were established by coinciding with the active period of wild mustard. We used a pressure-adjusted MATHABI-brand back sprayer with a fan jet nozzle under a constant pressure of three atm, and 300 liters of water per hectare was applied when plants had four to six leaves. The plants were checked regularly after watering and we evaluated the percent mortality and the dry weight of the plant at the end of the 28th day.

## Data analysis

After 28 days, plants in the dose–effect group were harvested and dried for 72 h at 65 °C to obtain the dry weight of the biomass. A dose–response analysis was performed using the R program and Log–logistic model below (*Ritz et al., 2015*):

$$y = c + \frac{d - c}{1 + \exp(b(\log(x) - \log(ED_{50})))}$$

In these formulas, Y is the response, d is the upper limit, c is the lower limit, b is the logarithmic (e) regression curve, and $ED_{50}$ is the dosage needed to reduce mass. The resistance factor and $ED_{50}$ values of the resistant populations were calculated using this formula.

## Molecular studies

### DNA extraction

Thirteen populations were found to be resistant to herbicides. Specimens from these populations were grown in the greenhouse and leaves were collected at the four-to-six leaf stage. The collected leaves were used to extract the genomic DNA with DNA extraction kit (*Danquah et al., 2002*).

### Polymerase chain reaction

The DNA isolated from the resistant thirteen populations were used in PCR studies to amplify Trp-574 residue with the ALS3F (5′-GGRGAAGCCATTCCTCC-3′) and ALS3R (5′-TCARTACTWAGTGCKACCATC-3′) primer sets (*Tan & Medd, 2002*). About 25 μl PCR reaction were prepared as follows: 10X PCR buffer, 70 ng genomic DNA, 0.50 mM primer, 3.5 mM MgCl$_2$, 0.9 mM DNTP, and 1 U Taq DNA Polymerase (Thermoscientific, Vilnius, Lithuania). The PCR program was set to: denaturation 94 °C 2 min, 60 °C 30 s, 72 °C 1 min, 35 cycles 94°C 20 s, 59 °C 30 s, and 72 °C 1 min; the melting temperature was increased 1 °C per cycle, and was 72 °C 5 min and 4 °C 1 min by the end (*Tan & Medd, 2002*). The amplification products were separated on a 1% agarose gel in 1 × TAE buffer.

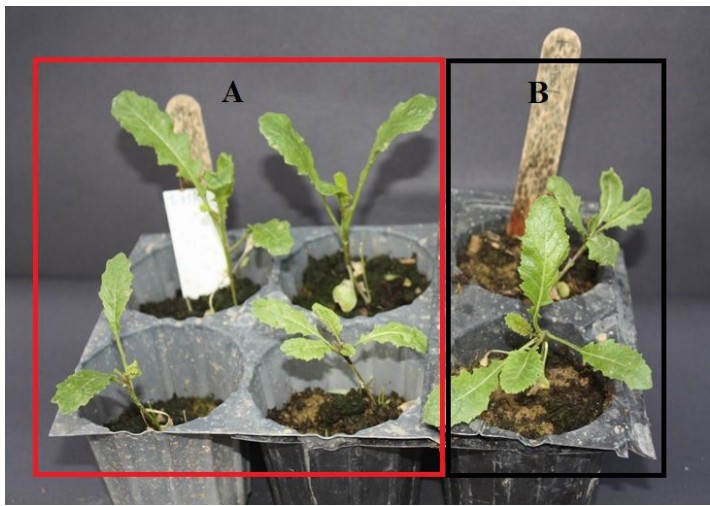

**Figure 2** **Population showing resilience as a result of bioassay study.** As a result of N dose administration during the Bioassay study. (A) Plants treated with N doses, (B) Untreated control plants.

### DNA sequencing

PCR products were sequenced in the REFGEN laboratory (Ankara) and the obtained sequences were subjected to blast comparison with *Arabidopsis thaliana* L. and other ALS-resistant weeds published in NCBI database.

## RESULTS

### Results of bioassay studies

We screened a total of 310 populations collected from Amasya, Çorum, Tokat and Yozgat for bioassay studies. Thirteen populations with less than 80% active compound controls were selected as being resistant and were used in dose–response studies (Fig. 2).

### Result of dose response studies

Tribenuron-methyl was applied at doses two, four, eight and 16-fold apart from the applications with registered doses. The $ED_{50}$ values of the populations and their resistance factors are shown in Table 1. The highest and lowest rates were 7.2 (A-007) and 1.03 (T-034), respectively.

### Trp-574 gene region base sequence analysis

The sequences of 13 resistant populations were compared with susceptible (X15514) Arabidopsis and susceptible wild mustard populations (AY964648). We downloaded FJ655876, FJ655877, AY964658, AY964661 and AY964663 from NCBI to determine their mutation-resistant accessions; these were used for comparison and to observe the mutation points. The presence of some nucleotide differences these base differences were not significant and only T-036 (MW252058) showed target site mutation (Trp-574-Leu) resistance.

**Table 1** ED$_{50}$ values and resistance index of wild mustard populations against tribenuron methyl.

| Population Number | Estimated ED$_{50}$ | Standard Eror | ED$_{50}$ Lower Limit | ED$_{50}$ Upper Limit | Resistance Factor |
|---|---|---|---|---|---|
| A-007 | 7.157 | 0.873 | 5.986 | 8.625 | 7.2 |
| A-022 | 3.891 | 0.360 | 3.164 | 4.618 | 3.9 |
| C-017 | 6.894 | 0.621 | 5.146 | 8.011 | 6.9 |
| T-010 | 3.927 | 0.145 | 2.986 | 4.863 | 3.9 |
| T-011 | 4.251 | 0.287 | 3.881 | 4.914 | 4.3 |
| T-012 | 7.656 | 0.544 | 5.719 | 9.271 | 7.7 |
| T-034 | 1.029 | 0.102 | 0.837 | 1.405 | 1.0 |
| T-036 | 2.141 | 0.349 | 1.452 | 2.992 | 2.1 |
| T-038 | 3.213 | 0.405 | 2.067 | 4.118 | 3.2 |
| T-050 | 5.444 | 0.109 | 5.187 | 5.972 | 5.4 |
| Y-021 | 3.168 | 0.857 | 1.546 | 4.639 | 3.2 |
| Y-022 | 2.878 | 0.382 | 2.108 | 3.649 | 2.9 |
| Y-026 | 2.330 | 0.578 | 0.164 | 3.495 | 2.3 |

**Notes.**

ED$_{50}$, Effective dose %50.

The susceptible population standard is taken as 1.

| | 570 Met | 571 Val | 572 Met | 573 Gln | 574* Trp | 575 Glu | 576 Asp | 577 Arg | 578 Ser |
|---|---|---|---|---|---|---|---|---|---|
| X51514.1 *A. thaliana* | ATG | GTT | ATG | CAA | TGG | GAA | GAT | CGG | TTC |
| Y-021 | ATG | GTT | ATG | CAA | TGG | GAA | GAT | CGG | TTC |
| T-036 | ATG | GTT | ATG | CAA | TTG | GAA | GAT | CGG | TTC |
| Control | ATG | GTT | ATG | CAA | TGG | GAA | GAT | CGG | TTC |
| T-003 | ATG | GTT | ATG | CAA | TGG | GAA | GAT | CGG | TTC |
| A-022 | ATG | GTT | ATG | CAA | TGG | GAA | GAT | CGG | TTC |
| T-034 | ATG | GTT | ATG | CAA | TGG | GAA | GAT | CGG | TTC |
| T-010 | ATG | GTT | ATG | CAA | TGG | GAA | GAT | CGG | TTC |
| T-038 | ATG | GTT | ATG | CAA | TGG | GAA | GAT | CGG | TTC |
| Y-092 | ATG | GTT | ATG | CAA | TGG | GAA | GAT | CGG | TTC |
| Y-026 | ATG | GTT | ATG | CAA | TGG | GAA | GAT | CGG | TTC |
| T-050 | ATG | GTT | ATG | CAA | TGG | GAA | GAT | CGG | TTC |
| T-012 | ATG | GTT | ATG | CAA | TGG | GAA | GAT | CGG | TTC |
| T-011 | ATG | GTT | ATG | CAA | TGG | GAA | GAT | CGG | TTC |
| T-014 | ATG | GTT | ATG | CAA | TGG | GAA | GAT | CGG | TTC |
| A-007 | ATG | GTT | ATG | CAA | TGG | GAA | GAT | CGG | TTC |
| C-017 | ATG | GTT | ATG | CAA | TGG | GAA | GAT | CGG | TTC |
| Y-022 | ATG | GTT | ATG | CAA | TGG | GAA | GAT | CGG | TTC |
| AY964663.1_Sinapis_arvensis Theodore | ATG | GTT | ATG | CAA | TTG | GAA | GAT | CGG | TTC |
| AY964661.1_Sinapis_arvensis | ATG | GTT | ATG | CAA | TGG | GAA | GAT | CGG | TTC |
| AY964658.1_Sinapis_arvensis | ATG | GTT | ATG | CAA | TGG | GAA | GAT | CGG | TTC |
| AY964648.1_Sinapis_arvensis | ATG | GTT | ATG | CAA | TGG | GAA | GAT | CGA | TTC |
| FJ655877.1_Sinapis_arvensis_MRS | ATG | GTT | ATG | CAA | TGG | GAA | GAT | CGG | TTC |
| FJ655876.1_Sinapis_arvensis_KNF1 | ATG | GTT | ATG | CAA | TGG | GAA | GAT | CGG | TTC |

**Figure 3** **Comparison of the base sequence of resistant samples with different Sinapis arvensis samples.** Nucleotide sequence of the acetolactate synthase (ALS) gene from tribenuron-methyl (TM) susceptible and TM-resistant *Sinapis arvensis* plants collected from winter wheat fields in TOKAT province, Turkey. Codons indicate a change at the 574 position from TGG (tryptophan=Trp) to TTG (leucine=Leu), associated with conferring cross-resistance to ALS inhibiting herbicides.

The base difference was detected from the results of the mutation during amino acid synthesis. The order of amino acids at Trp-574 was determined to be TGG instead of TTG. Thus, we studied the transformation of the tryptophan amino acid (W) transformation to leucine (L). The sequence comparisons of resistant and susceptible populations are shown in Fig. 3.

# DISCUSSION

Weed infestation and competition are major factors causing reduced growth and significant yield losses for wheat production. Herbicides offer cheaper and more effective chemical control for wheat fields, however the intensive use of herbicides may increase the risk of producing resistant weeds. Herbicide resistance is expressed as the ability of weeds to stay alive after continued exposure to low doses of herbicides. Resistance has been studied in previous studies and the results determined that weed resistance emerged after regular applications of ALS-inhibiting herbicides over three to five years. (*Wang et al., 2019*; *Powles & Yu, 2010*; *Tranel & Wright, 2002*). We investigated the ALS herbicide resistance level of wild mustard in 310 wild mustard populations; only 13, including seven from Tokat, three from Yozgat, two from Amasya and one from Çorum, developed resistance.

Herbicides are generally absorbed by the leaves and roots of plants and must be bound to the enzyme to become active. Acetolactate synthase is one of the enzymes involved in herbicide activation and this function is inhibited by herbicides including imazamox and tribenuron methyl (*Shaner & O'Connor, 1991*). Resistance emerges when a nucleotide difference occurs due to an amino acid point mutation at the binding site (*Kolkman et al., 2004*). There are 17 amino acids, to date, upon which substitution in the ALS gene confers ALS-inhibitor resistance in plants, yeast, bacteria, and green algae (*Christoffers et al., 2006*; *Duggleby and Pang, 2000*; *Duggleby et al., 2008*; *Tranel & Wright, 2002*). Although mutations in 17 amino acid have been reported in plants, bacteria, yeast, and green algae, only eight amino acid positions, including Ala 122, Pro 197, Ala 205, Asp 376, Arg 377, Trp 574, Ser 653 and Gly 654, have been reported in weeds (*Li et al., 2008*; *Yu and Powles, 2014*; *Powles & Yu, 2010*; *Gherekhloo et al., 2018*). Each amino acid plays a role in the resistance of different herbicide groups. Gly654 develops resistance to IMI, SU, and SCT; Pro197 is involved in the resistance to TP and SU; Ser653 induces resistance to PTB and IMI; Ala122 and Ala205 cause resistance to IMI; Asp376 and Trp-574 leads to the resistance in all ALS herbicide groups (*Cruz-Hipolito et al., 2013*). Resistance to sulfonylurea and imidazolinone develop after the mutation of Trp-574 to Leu, while resistance to tribenuron-methly occurs following Trp-574 to the Gly point mutation (*Warwick, Sauder & Beckie, 2005*; *Jian et al., 2011*).

Mutations in codons 197 and 574 are considered to be important points for ALS inhibitors (*Powles & Yu, 2010*; *Tranel et al., 2019*; *Zhao et al., 2020*). Trp-574-Leu has the ability to develop resistance to all groups of ALS inhibitor herbicides (*Pandolfo et al., 2016*). The Trp-574 Leu mutation occurs with a higher level of resistance against IMI (Imazamox) and SU (Tribenuron methyl), according to various studies (*Tranel and Wright, 2002*; *Warwick, Sauder & Beckie, 2005*; *Tan & Medd, 2002*; *Rosaria et al., 2011*; *Christoffers et al., 2006*; *Deng et al., 2017*; *Ntoanidou et al., 2017*; *Ntoanidou et al., 2019*).

We screened 13 resistant populations for dose effect studies, and the Trp-574 amino acid mutation and base differences were observed in the T-036 population collected from a wheat field in Tokat that had been sprayed with imazamox, and tribenuron-methyl. The Blast results of this population indicated a nucleotide difference at the Trp-574 amino acid and showed TGG transversion to TTG. *Gherekhloo et al. (2018)* conducted a similar

study in Iran to determine the cross resistance against tribenuron-methyl in wild mustard and observed base differences in the nucleotides. The base order at Trp-574 was TTG instead of TGG in AL-3, G-5, and Ag-Rs populations. *Ntoanidou et al. (2017)* determined the mutation of Trp-574 of nine populations after tribenuron-methyl and imazamox were applied in Greece.

Studies in Turkey to assess the resistance to ALS inhibiting herbicides were conducted at a bioassay level (*Avcı, 2009*; *Aksoy et al., 2010*; *Uygur et al. 2014*; *Gürbüz, 2016*). One study to determine the molecular detection of chlorsulfuron resistance of wild mustard was conducted but did not find the point mutation (*Topuz, 2007*). *Kaya-Altop (2012)* collected plants from paddy fields and investigated the resistance of *Cyperus difformis* L. against ALS inhibitors. They determined that there was a point mutation in Pro 197 amino acids. (Erken, 2016) screened *Lolium* spp. against multiple resistances and observed mutations in the Pro 197 amino acid, however, Trp-574 mutations were not reported. *Erken (2016)* also investigated *Lolium* spp. against multiple resistances and only observed mutations in the Pro 197 amino acid; Trp 574 mutations were not reported.

Our study is the first to show the Trp-574 amino acid mutation in *S. arvensis* in wheat fields in Turkey. Our results will contribute to future studies and will assist in the selection of proper management methods.

## CONCLUSIONS

In Turkey, *S. arvensis* with the ALS inhibitors were found to develop resistance to herbicides. The active ingredient of tribenuron-methyl used in wheat growing areas was found to have no effect on *S. arvensis*. Due to this mutation, the Leu transformation of Trp 574 occurred at the ALS enzyme, resulting in mutational resistance. Producers in the region where T-036 is grown regularly use herbicides with tribenuron-methyl and imozamox. We identified the Trp 574 mutations in the wild mustard plant for the first time in Turkey. This manuscript was prepared as a part of PhD thesis conducted in Gaziosmanpaşa University.

### Funding
This research was funded by Tokat Gaziosmanpaşa University Scientific Research Center (BAP-2018/27). The funders had no role in study design, data collection and analysis, decision to publish, or preparation of the manuscript.

### Grant Disclosures
The following grant information was disclosed by the authors:
Tokat Gaziosmanpaşa University Scientific Research Center: BAP-2018/27.

### Competing Interests
The authors declare there are no competing interests.

## Author Contributions

- Bahadir Sin conceived and designed the experiments, performed the experiments, analyzed the data, prepared figures and/or tables, authored or reviewed drafts of the paper, and approved the final draft.
- İzzet Kadıoglu conceived and designed the experiments, analyzed the data, authored or reviewed drafts of the paper, and approved the final draft.

## Data Availability

The data are available at NCBI Genbank: MW252058.

## Supplemental Information

Supplemental information for this article can be found online at http://dx.doi.org/10.7717/peerj.11385#supplemental-information.

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
