# Peer review of "Trp-574-Leu mutation in wild mustard (Sinapis arvensis L.) as a result of als inhibiting herbicide applications"

_PeerJ, doi:10.7717/peerj.11385_

## Round 0.1 · original submission · Major Revisions

As you can see, all reviewers raised some serious concerns. Therefore, please address their critiques and revise manuscript accordingly.

Reviewer 1 ·

Basic reporting

The authors report Trp-574-Leu MUTATION IN WILD MUSTARD (Sinapis arvensis L.) AS A RESULT OF ALS INHIBITING HERBICIDE APPLICATIONS in Turkey. However this is not the first time this resistance has been reported. Thus the manuscript has no novelty for PeerJ and should be rejected

Experimental design

The experiments are well designed

Validity of the findings

The results are valid and obtain from a well performed experiment with many samples

Additional comments

Reporting for ALS resistance due to Trp574 mutation has been reported before for Turkey

Reviewer 2 ·

Basic reporting

Figures are not appropriated. The authors should use a better map image (Fig 2), give the legends (Fig 3) and remove the software layout (Fig 4 and 5).
The English must be strongly reviewed specially in M&M section.

Experimental design

The investigatin was not rigourous due to other positions at ALS gene those can harbor target-site mutations. I my point of view, the authors sould investigate all ALS sequence to conclude about the remaining resistant populations.

Validity of the findings

As far as my concern, resistance factor must be calculate based on I50 values. 50% control or 50% mass reduction.

Additional comments

Congrats for your job.
However, there are some concerns about the methods and discussion.
- Why do not the authours investigated other target site mutations over the ALS gene?
- Why do you prefer showing ED90 instead of ED50. We know that ED50 is the best parameter to estimate the levels of resistance.
- The discussion must be improved. If you did not find Trp574Leu in the remaining resistant wild mustard populations, what would be the causes of resistance on those? I suggest you implement two or three more paragraphs in the discussion section talking about what it should change on weed control praticies at wheat fields in Turkey. Is there any other resistance mechanism evolved in wild mustard against ALS herbicides? If no, how about discuss around related species.

Annotated reviews are not available for download in order to protect the identity of reviewers who chose to remain anonymous.

Reviewer 3 ·

Basic reporting

Sinapis arvensis L. is a worst weed in wheat fields in Turkey. This study detected the level of resistance and investigate the TSR to tribenuron-methyl in Sinapis arvensis. These results are helpful for understanding the resistance status and the mechanism of Sinapis arvensis. to ALS inhibitors. However, there are also some shortcomings. The organization need to be improved some sentences about herbicide resistance description were also not rigorous. Therefore, I'm afraid that the authors should pay much attention to carefully improve the manuscript before it can be considered for publishing in Peer J.

Experimental design

no comment

Validity of the findings

no comment

Additional comments

The major concerns are as following:
1. The detailed information on the methods used for collecting the seeds is needed. Did the seeds bulk?
2. How many copies of ALS gene in Sinapis arvensis.
3. How about the ALS enzyme activity, suggest to supplement the In vitro ALS activity assay.
4. L151 "Unite Taq DNA Polymerase" is a high-fidelity thermostable DNA polymerase ?
5. The discussion section need to be significantly improved. In this section, based on the obtained results, the authors should compare the similarity and difference between this current research and other reports,and discuss the possible reasons, then make conclusion and pointed out the novelty and importance of this research.

Some of the spelling and/or grammar errors in the manuscript:
1. L25: Change "Tribenuron methyl" to "tribenuron-methyl".
2. L31: Italics for Latin names.
3. L48-49: Change to "75.6 billion dollars ", "2.5-2.7 billion dollars ", "5 billion dollars ".
4. L64: "As wheat is a hoe plant weed control" ?
5. L77-79: Check to the tense.
6. L95: Change to "Ala 122".
7. L96-97: Delete the sentence " se Ala-122……...(Powles and Yu, 2010; Vencill et.al., 2012) ".
8. L106: Change "have" to "had".
9. L113 "pre"?
10. L115: suggest change "gr da-1" to g a.i. ha-1".
11. L119: Change to "Do -responses to tribenuron-methyl"
12. L120-L121: Change to "1, 2, 4, 8 and 16-fold".
13. L129-130: "in %"?
14. L137: "%90"?
15. L150-151: Change to "0.5mM", "3.5mM", "0.9mM"."0.5 Unite Taq DNA Polymerase"?
16. L152: Change "sek" to "sec". Change "cycle" to "cycles".
17. L158-160: Check to the tense and italics for Latin names.

The English writing is somewhat poor. These are some tense and grammar errors here and there. I just correct some of these errors. The authors should ask for help from native English speaker to polish the manuscript to improve the quality and readability.

---

## Round 0.2 · Minor Revisions

Please address remaining concerns of reviewer #2, who pointed to several really minor issues.

Reviewer 2 ·

Basic reporting

No coments

Experimental design

No coments

Validity of the findings

No coments

Additional comments

Dear authors,
Congrats to improve your manuscript. The discussion section was really revised and well corrected.
Figures are now appropriated.
I have a few suggestions before make the manuscript accept.
- Change the description of ED50: I think welcome call it: Dose to 50% of mass reduction.
- I keep do not understanding "endurance or durability coefficient" in the Table 1. I believe it might be the "resistance factor" or "resistance index". If so, please specify in the Table which wild mustard population was your Susceptible standard

Once the authors make these minor revisions, the article can be considered welcome to publication!
Congratulations for your work and your phD project.

Sincerely,

Reviewer 3 ·

Basic reporting

no comment

Experimental design

no comment

Validity of the findings

no comment

Additional comments

no comment

---

## Round 0.3 · Minor Revisions

All remaining technical issues were addressed and the revised version is almost acceptable now.

Please have the manuscript proofed to resolve the language. PeerJ does not provide copyediting as a standard service, please ensure that the English language in this submission meets journal standards; this includes the use of clear and unambiguous text which is grammatically correct, and conforms to professional standards of courtesy and expression.

---

## Round 0.4 · accepted · Accept

Thank you for fixing the remaining linguistic issues and submission of the edited manuscript. It is acceptable now.